# Physical Activity and BMI before and after the Situation Caused by COVID-19 in Upper Primary School Pupils in the Czech Republic

**DOI:** 10.3390/ijerph19053068

**Published:** 2022-03-05

**Authors:** Jana Pyšná, Ladislav Pyšný, David Cihlář, Dominika Petrů, Lenka Hajerová Müllerová, Luděk Čtvrtečka, Anna Čechová, Jiří Suchý

**Affiliations:** 1Department of Physical Education and Sport, Faculty of Education, Jan Evangelista Purkyně University in Ústí nad Labem, České mládeže 8, 400 96 Ústí nad Labem, Czech Republic; jana.pysna@ujep.cz (J.P.); ladislav.pysny@ujep.cz (L.P.); david.cihlar@ujep.cz (D.C.); dominika.petru@ujep.cz (D.P.); l.ctvrtecka@seznam.cz (L.Č.); 2Department of Pedagogy, Faculty of Education, West Bohemia University in Pilsen, Chodské náměstí 1, 306 14 Pilsen, Czech Republic; hajerova@kpg.zcu.cz; 3Department of Didactics, Faculty of Physical Education and Sport, Charles University in Prague, José Matího 31, 162 52 Prague, Czech Republic; cechovaannaa@gmail.com

**Keywords:** physical activity, body mass index, COVID-19, pupils, health, lifestyle, sport, obesity

## Abstract

Regular physical activity is a very important factor in the healthy development of an individual and an essential part of a healthy lifestyle. However, today’s population still suffers from an insufficient amount of exercise caused mainly by technological progress and often inappropriate conditions for practising sports. In relation to this, we are grappling with a steady increase in obesity. During the COVID-19 pandemic, conditions for regular physical activity became even more unfavourable, with the declaration of a state of emergency and antipandemic measures leading to the closure of sports grounds and sporting competitions. Using a questionnaire survey of a sample of children (*n* = 1456), we found that, already before the pandemic, 69% of the observed sample had not met the recommended amount of physical activity, and only 67% of the sample was of normal weight. By comparing both groups after the end of pandemic restrictions, we found statistically significant differences at examined indicators of the children’s Body Mass Index (BMI), their physical activity, and free time spending habits. We noticed the significant differences in BMI indicators in two different categories, normal weight (7.5%) and stage 1 obesity (1.66%). Simultaneously, we noticed differences in the children’s physical activities, especially with children who attend sports playgroups connected to athletic development (8.74%). More differences were noticed in free time spending habits indicators; the most significant ones were with the children who spend their free time behind the personal computer for more than 14 h a week (5.4%) and with the children who spend their free time on social media for 8–14 h a week (18.56%).

## 1. Introduction

According to Perič and Březina [1], exercise and sport are a fundamental part of every individual’s life, an essential manifestation of a healthy lifestyle that is indispensable for the proper development of every individual [2]. According to the World Health Organization [3], physical activity includes any activity produced by skeletal muscles causing an increase in pulse and respiration rate. It includes habitual physical activity, controlled performance, competitive physical activity with the aim of best performance, and leisure-time physical activity [4]. Sport and physical activities are very important components of a healthy lifestyle and vital factors influencing health. Spontaneous physical activity is the most significant component of exercise in children, where appropriate development of movement abilities and acquisition of movement skills occurs, thus contributing to the course of further development of the child and anchoring interest in physical education and sports activities [5].

School physical education is also an integral part of children’s exercise, but it cannot cover the real exercise requirement. In particular, school physical education helps break the stereotype of mostly sedentary teaching and creates important movement patterns in children. Undoubtedly, sports clubs can be included among pupils’ common sports activities, intended to mainly focus on performance [6,7,8,9]. The formation of positive relationships and attitudes towards physical activity, as well as their lifelong application, occurs during childhood [10]. Recommendations for children and adolescents aged 7 to 15 years state the practice of moderate-intensity physical activity for at least 60 min per day. Under these conditions, physical activity accounts for about 30–50% of an individual’s daily energy expenditure and is one of the factors contributing to a positive energy balance, by decreasing energy and fat intake, and the prevention of lifestyle diseases [7,10,11,12]. The European College of Sports Medicine, the US National Institute of Health, and the American College of Sports Medicine recommend a volume of physical activity with an energy expenditure of 4200 kJ/week as sufficient for beneficial health effects [13,14,15,16,17].

Regular physical activity increases physical fitness, lowers cholesterol levels, and appears to be the most effective factor in preventing lifestyle diseases by contributing to mental freshness and resistance to stress, improving blood circulation and brain oxygenation, and preventing muscle pain and the development of functional musculoskeletal disorders and chronic noninfectious diseases [18,19,20]. In the same context, engaging in sports may help prevent serious cardiovascular, metabolic, and cancer diseases and improve the quality of life. For example, engaging daily in a suitable level of physical activity may contribute to weight reduction and weight maintenance and improve the metabolic complications that usually accompany obesity [21,22].

In the development of an individual from childhood to adulthood, there is currently a significant decline in the amount of physical activity, mainly due to a sedentary lifestyle. In this context, it is necessary to remember that the human organism, as a result of evolution, is set up to be exposed to relatively intense physical stimulation for the natural and healthy development of the individual [14]. In early childhood, a sedentary lifestyle is only minimally prevalent in approximately 8% of girls and 6% of boys and is preferred by 25% of women and 22% of men by the age of 20. In developed countries, this is mainly due to a change in leisure time use in favour of less physically demanding activities. This can lead to physical inactivity over time [17,23,24]. Bunc [25] reports that energy intake has been stagnating or decreasing over recent decades in the Czech Republic and Central European countries. However, on the other hand, energy expenditure is also decreasing significantly.

According to Rychtecký and Tilinger [26], period of adolescence is characterised by risk behaviour manifesting reduction in the amount of physical activity. The changes characteristic of adolescence (e.g., psychological, hormonal, behavioural changes) with the persistent trend of decreasing the amount of physical activity, which has arisen due to the modernisation of society and the constant advancement of information technology, may negatively affect the lifestyle of adolescents [14]. The result, for example, is the abovementioned ever-increasing prevalence of obesity, which poses a serious problem for our society, with the representation of overweight and obese people approaching sixty per cent [27]. Children and young people are also at risk. The assessment of the development in the second half of the twentieth century already showed a long-term positive trend of increase in body weight of children and youth [5,28,29,30,31]. Therefore, we are talking about childhood obesity, which is conditioned by a change in the lifestyle of children with a characteristically significant limitation of their spontaneous physical activity. Thus, at the same time, obesity becomes a large-scale, not just economic, problem for the entire healthcare system [32,33].

In addition to the aforementioned health risks of insufficient physical activity in adolescent children, the impulsivity, irritability, aggression, reduced ability to concentrate, and reduced self-control are also common. The former experience of adventure in a variety of children’s physical games and activities is nowadays replaced by a virtual experience with simultaneous minimal physical activity [34,35]. All of these health risks increase proportionally with Body Mass Index (BMI) values [21,22]. BMI is one of the basic indicators of body weight in connection to overweight and obesity. It is a globally recognised index, applicable to all age categories. Weight ranges, together with the resulting health risks, are provided for the respective age groups [16,36].

A number of experts have pointed out the ongoing need to develop appropriate physical activity intervention programmes that promote regular physical activity in children and adolescents, which, especially during childhood, significantly affects physical and mental health [14,37,38]. Moreover, these patterns and exercise habits are carried on by the child into adulthood. Physical activity is recommended by experts not only in the Czech Republic but also worldwide as an important component in the fight against the increasing prevalence of obesity, and its promotion has become a national health priority. The findings of several studies show that a large proportion of older children does not participate in the recommended daily sixty minutes of moderate to vigorous physical activity, so it is necessary to intervene and change their behaviour [15,29,34,39]. Experts emphasise that appropriate physical activity interventions can potentially increase children’s level of physical activity and, moreover, can be included in school health programs. The focus on youth physical activity has led to the development of guidelines and recommendations specific to adolescents, children, and infants [14,40].

Exercise intervention should take into the account a range of indicators, such as health status, exercise history, the current level of fitness, and physical ability. The actual design of physical activity should include the form, intensity, duration, frequency, instructions, and checking of the effect [20,41]. It is important to respect the child’s developmental stage, i.e., anatomical, physiological, psychological, and training specificities. In the first stage of the designed exercise regimes, it is necessary to focus on the cultivation of movement activities. The load should not last more than 40 min with pulse rate (PR) values of 80–90% of the maximum PR. Exercise programmes should primarily load large muscle groups and should include compensatory exercises. Under these conditions, exercise has been shown to improve health, positively promote a healthy lifestyle, and contribute to a higher quality of life in various ways. It improves psychological well-being improves self-image, and also brings pleasurable emotional experiences [26].

On 12 March 2020, the Government of the Czech Republic declared a state of emergency due to the health threat from the rapid spread of the SARS-CoV-2 coronavirus, which lasted until 30 June 2020 [42,43]. Under the declaration of a state of emergency, certain human rights and freedoms may be restricted, pursuant to Crisis Act No. 240/2000 Coll. On the basis of the declaration, new government measures were imposed, including the prohinition of sporting events, in person school attendance and educational events, as well as the free movement of the population; the state of emergency thus affected, among others, the area of education, organised sport and leisure-time physical activities [44]. On 1 September 2020, a state of emergency was again declared by the Government of the Czech Republic and lasted until 14 February 2021 [45].

The Ministry of Health first established basic hygiene rules for schools and other facilities. In the spring, from 12 March 2020–30 June 2020, the Government announced new measures on school attendance and educational events, banning extracurricular activities, competitions, and students’ access to school facilities. Regulations were still in force that further restricted the operation of school facilities, in particular physical education, organised sports activities and sports competitions [46]. On 2 November 2020, a government resolution newly prohibited the presence of pupils and students in classrooms and sports activities [47]. From 30 November 2020 until June 2021, the regulations that further restricted the operation of school facilities, in particular, physical education, organised sports activities, and sports competitions, were again in force [48,49]. The government measures restricted, among other things, the free movement of persons, education, organised physical activities, and participation in domestic and international competitions.

The above summary of the government measures shows that the COVID-19 pandemic has spread globally and that the resulting bans and restrictions aimed at limiting the spread of the disease have also significantly impacted children’s physical activity. The closure of school gymnasiums, sports halls, swimming pools, public parks and playgrounds, and a range of other sports venues was associated with a significant reduction in school-based physical education, organised sports, and leisure-time physical activity for children. All this has resulted in an increased amount of time spent at home without the usual opportunity for regular sporting activity. During the harshest government measures, even movement outside one’s own home was restricted [48].

According to Bates et al. [50] and Yang et al. [51], the impact of antipandemic measures has thus caused a reduction in social contact that has had far-reaching adverse effects on the mental and physical health of children and adolescents, exacerbated the current problem of low levels of physical activity, and increased the prevalence of sedentary lifestyles. In their study, Dubuc et al. [52], Moore et al. [53], and Velde et al. [54] found a negative impact of government measures on the exercise and play behaviours of Canadian, Dutch, and other children and adolescents. Specifically, during the initial lockdown period these groups were less active, played outdoors less, spent more time in sedentary activities, increased their interest in PC or television, and slept more compared with the period before the restrictions. Similar results reporting a decrease in physical activity among children in Shanghai during the emergency were found by Xiang et al. [55]. The authors also reported a significant increase in the number of physically inactive students and an increase in the average weekly time spent in front of a television screen. Another cross-sectional study conducted among adolescents in two European countries (Italy and Spain) and three Latin American countries (Brazil, Chile, and Colombia) reported changes in the amount of physical activity and the consumption of industrially processed foods. A decrease in physical activity and increased consumption of industrially processed foods were observed in the probands, with more pronounced changes found in Latin American adolescents, as reported by Ruíz-Roso et al. [56]. The impact of reduced physical activity on psychosocial factors in adolescents has been investigated by Slimani et al. [57]. In their research, they found impaired behaviour, poorer mood, a decline in psychological fitness or socialization problems.

## 2. Materials and Methods

The research survey was conducted following the approval of the Ethics Committee of the Faculty of Education UJEP (1/2019/01) in 15 primary schools with the consent of school principals and the informed consent of parents. The survey was carried out in two stages. The first stage was conducted before the outbreak of the COVID-19 pandemic, from April 2019 to February 2020 (1133 pupils) in 15 primary schools in North Bohemia in the Czech Republic. The second stage took place after the stringent COVID-19 pandemic counter measures ended, from September 2020 to January 2021, with 323 pupils from 5 of the original 15 primary schools included in the survey, whose parents provided written consent to participate in the research. Between the different stages of data collection, anti-pandemic measures were adopted in the Czech Republic, which, among other things, severely limited children’s sport and physical activity. The survey attempted to determine whether there were any differences in physical activity and BMI, pre- and post-COVID-19 crisis, among the upper primary school pupils in North Bohemia.

The sample monitored consisted of 1456 pupils (mean age 12.9 years) of primary schools in North Bohemia. Of these, 681 were girls, and 775 were boys (Table 1). The average age of the girls was 12.7 years, and the average age of the boys was 13.1 years. The sample consisted of upper primary school pupils from 15 schools in North Bohemia. The schools were selected using a group selection with the definition of the strata characterizing stratified selection. These schools met the requirements for the stratification of schools in terms of location in the region and, at the same time, met the characteristics of the environment of North Bohemia. For the entire sample of 1456 pupils, the level of lifestyle was ascertained using a questionnaire survey, anthropometric data, and body composition assessed by BMI. Since no statistically significant difference was found between girls and boys in the indicators studied, we present the population without gender distribution.

The methods used include somatometry to measure basic anthropometric indices (age, body weight, and height). To assess body composition, we used the Body Mass Index, and to ascertain the level of lifestyle, we applied the questionnaire method using the CAV (National anthropological research) 2001 questionnaire for children and adolescents [58]. For the purpose of this research, we used questions within Sets I, II, and III.

For measuring the basic anthropometric indicators, we used a calibrated scale with accuracy to one decimal place (measurement values are given in kilograms (kg)) and calibrated measuring device with accuracy to two decimal places (body height values are given in centimetres (cm)).

For measuring the body composition, we used the calculation of the weight ratio in kilograms to the square of the height in meters. To evaluate the observed results, we relied on the BMI-for-age evaluation [59].

The CAV (National anthropological research) 2001 standardised questionnaire for children and youth, Vignerová and Bláha [58], contains a total of 15 closed questions in 5 sets. Set I, identification questions—age, sex, height, weight, and nationality; Set II, physical activity—“Where do you do sports most often?”, “How many hours a week do you do sports outside Physical Education (PE) at school?”, “Do you own a bicycle and use it as a means of transport or for sports?”; Set III, leisure time—”How many hours a week do you watch television? “, the question “How many hours a week do you work on a computer, play computer games or Play Station games?”, “How many hours a week do you spend on your mobile phone or communicating on social networks?”; Set IV, eating and drinking habits—”Do you eat breakfast in the morning? “, “Do you eat snacks at school?”, “Do you have a hot meal at school?”, “Do you eat snacks in the afternoon?”, “Do you have dinner?”, “Do you eat fruit and vegetables regularly?”; Set V, taking care of the body habitus—weight tracking, dieting.

For statistical methods of data processing and evaluation, we used descriptive statistics significance tests (chi-square test of independence). We used statistical significance tests to demonstrate the relationships under study. Null hypotheses were rejected with less than a 5% probability of error, i.e., when our *p* value (probability of error in rejecting the null hypothesis) fell below 0.05 [60].

## 3. Results

The results presented in Table 2 show a statistically significant difference in BMI values between the study groups. During the evaluation of BMI, it is clear that after restrictions (second phase of the research), the biggest difference can be seen in the BMI of the children with normal weight and BMI of either underweight or the stage 1 obesity children.

Table 3 presents the distribution of pupils according to the answers to the question “Where do you do sports most often?” before and after the situation caused by the COVID-19 pandemic. From these answers, it is obvious that the most significant difference is to be seen in the group of children regularly attending sports clubs with competitive training, followed by children freed from school physical education and the group who do/play sports during after-school activities with no athletic physical training. More differences were noticed with the children regularly doing/playing sports with their families, friends, or at school. The difference between the groups in the indicator of participation in physical activity at the first and second data collection is statistically significant.

Table 4 presents the number of hours spent doing sports in the study sample before and after the situation caused by the COVID-19 pandemic. The amount of recommended minimum sports activity is based on the recommendations of doctors, who say that sports activity in children should take at least 1 h a day. Based on this, we divided the children into two groups. The group doing sports are children who meet the recommended amount of sports activity (at least 7 h per week); the group not doing sports are children who do not meet this recommendation (less than 7 h per week). The results of the comparison of the observed indicator at the first and second stages of the data collection process are not statistically significant; despite this, we can see the difference with the physically active children.

Table 5 shows the results of children’s answers to the question “Do you own a bicycle and use it as a means of transport or for sport?” During the comparison period (before and after the situation caused by the COVID-19 pandemic), there is a statistically significant difference with the children who use bikes as a means of transportation.

Table 6 shows the results of pupils’ responses to the question “How many hours a week do you watch television?” before and after the situation caused by COVID-19. From the listed values, it is clear that after the situation caused by the COVID-19 pandemic, the biggest difference is with the children who watch television regularly for 4–7 h a week. The comparisons of the results of the responses between the groups at the first and second stages of the research are not statistically significant. 

Table 7 shows an evaluation of the probands’ answers to the question “How many hours a week do you work on a computer, play computer games or Play Station games?” The results indicate the biggest differences among the children, who spend more than 4 h a week behind the computer and in the interval of 14+ hours a week during the COVID-19 pandemic. These results, which compare probands’ responses before and after the situation caused by COVID-19, are statistically significant.

Table 8 shows the results of spending time on the mobile phone or communicating on social networks. After COVID-19, we observed the most significant difference in the time interval of 14+ a week, 0–3 h a week, and 4–7 h a week spent on mobile and social media. These results are statistically significant.

## 4. Discussion

The entire study sample of upper primary school children in North Bohemia in the Czech Republic (*n* = 1456) was surveyed using a questionnaire to obtain information on leisure activities and BMI. We attempted to ascertain the focus of the pupils’ extracurricular physical activity and the average time they devote to it per week. For further elaboration, the variable of out-of-school sports activity was categorised (with doing sports and not doing sports). An alternative was created showing pupils reporting at least 7 h of physical activity per week, which corresponds to the condition “at least one hour per day on average”, and pupils reporting less than 7 h per week. Here, we relied on expert claims that cite 7 h of sports activity per week as the optimal amount [25,61]. We also investigated the amount of time pupils spend watching television or using the computer and the amount of time they spend on social media. In particular, we focused on comparing these indicators ascertained in the first and second stages of data collection, i.e., before and after the announcement of antipandemic measures that closed sports venues and sports competitions for half a year during the COVID-19 disease pandemic. In the first stage of the research, we found that 69% of pupils did not meet the recommended amount of sporting activity in their leisure time. In the second stage of data collection, we detected a 5% difference among the children. Thus, we confirm the claims of Sigmundová and Sigmund [4], who state that the physical activities of children and adolescents have changed significantly over the last 10 to 20 years. Extracurricular activities of young people have been reduced to watching television, communicating via the internet, and gathering in groups which, unfortunately, do not always have an appropriate focus. There are fewer young people playing sports who are willing to exert more physical effort and overcome obstacles of various types. We draw on the results of Jíra [62], where young people reported watching television as their favourite leisure activity. It is also interesting to note the results of the STEM/MARK (a full-service agency specializing in market research) survey, which brought information that children spend an average of 11 h a week in front of the television, 5 h and 20 min on the computer, 5 h in unorganised activities and only 2 h in after-school clubs [63]. At the same time, concerning the abovementioned issue, it is necessary to support the opinion of the National Health Institute in Prague [64], stating that sport in schools is the most widely available resource for promoting sports activity among young people and it is therefore worth considering to increase the amount of sports activity by expanding the hourly allocation of sports subjects in schools.

The decline in the amount of sports activities is also related to the period of puberty itself. This can be explained by the assumption of a group identity, where it is the group that provides support and shares interests and concerns. In contrast, the parent, who has often been the main supporter of the child in a sport, thus takes a back seat as a role model. The child often prefers other interests, such as the conformity associated with the style of speech, dressing or behaviour in exchange for time spent playing sports. It is therefore very important to further shape, encourage and motivate the relationship with physical activity at this age [29]. It is important to use the influence of the family in creating a positive relationship with physical activity, especially in the preschool period. Subsequently, the school and the influence of teachers should certainly be added during the school attendance period. Both entities have a clear and full responsibility for the formation of a positive attitude towards sport and a healthy lifestyle. These results confirm the negative impact of the six months of isolation of children from sporting activities associated with an increase in the number of hours spent on personal computers and social networks [65]. 

Alarming results were observed in the evaluation of the results in the second stage of data collection. i.e., after the situation caused by the COVID-19 disease, when there was a significant difference with the children who do/play sports in clubs with competitive training and with the children who are freed from school physical education. At the same time, there is a difference with the children who stated that they do/play sports only in after-school activities with no athletic physical preparation and with the children doing/playing sport with their friends and families.

Our results confirm the claims of [29], who cite the age of the children as a possible factor influencing the increasing number of children playing sports with friends or family. It is likely that there was also a transfer of habits from the period when sports venues and sports competitions were closed, and the family had the opportunity to spend more free time together and actively. The significant reduction in the number of children playing sports in clubs with competitive training during the second data collection after the period of antipandemic measures was probably amplified by the influence of risky behaviour of children during adolescence. According to Pyšná et al. [66], puberty is considered the most dynamic transformation in an individual’s life, affecting all components of an individual’s personality. Pubescents show increased excitability, moodiness, absentmindedness, and affective reactions. They strive for independence, and increasing criticalness is the reason pupils are no longer willing to perform their duties without objection and very often display defiance and negativity, including in the sporting environment; they very often leave or move to after-school clubs without competitive sports training. In our further research, we focused, in particular, on comparing the time pupils spend watching television or using a personal computer and how much time they spend on social media. In the results for the time spent watching television, we found that after the situation caused by COVID-19. i.e., during the second stage of the research, there was a difference among the children who spent their free time in front of the television, PC, or on social media. The biggest difference came with the children who spend 14 or more hours a week on social media. 

These results are generally due to a preference for interests during adolescence, and at the same time, there is certainly a negative impact of six months of sporting isolation and spending time at home during online learning. According to Forýtková and Bourek [67], girls are more likely to spend their free time with friends or in hobby groups, but unfortunately, without sufficient physical activity, while boys have a closer inclination towards gaming technology and PC work. According to Pyšná et al. [39], it is evident that with increasing age there is an increase in the time spent communicating on social networks, with social networks influencing many areas of people’s lives. The impact of virtual environments is being researched and discussed by experts across scientific disciplines. Given that social networking is part of the contemporary modern life of adolescents, we believe it is appropriate to share posts related to a healthy lifestyle. According to the results, which confirmed that pupils use Instagram as a source of information, it can also be recommended that professionals start actively using social networks and that more accounts with verified professional information are created. For example, content on Instagram could be rated by doctors and nutritionists so that a young person can distinguish between accurate and false information according to the tags assigned. When evaluating the BMI in our research survey, we can confirm that there has been a steady increase in overweight and obesity in our children.

This trend follows a number of Czech and international studies. e.g., Pyšná et al. [20], Kunešová et al. [29], World Health Organization [59], National Institutes of Health in Prague [64], Caldeira et al. [68], Flegal et al. [69], Canadian Community Health Survey [70], Kubínová et al. [71], and Pyšná et al. [72]. In assessing the trend, it should be noted that already during the first stage of the research, we observed a relatively high proportion of children (33%) with other than normal weight. During the second stage of data collection, i.e., after the measures connected with the pandemic COVID-19, we observed the difference among the children in the normal weight category and those who were underweight or in stage 1 obesity.

The difference between the groups in the BMI indicator during the first and second data collection is statistically significant. During the first data collection, we observed 67% of children with optimal weight. These differences we observed at the end of pandemic measures can be compared to the results of the study by Pyšná et al. [20]. From the number of obese children in our study, we can assume that changing our children’s lifestyles will lead to further serious health problems, such as metabolic changes in the body associated with increasing weight and obesity, significantly reducing their future quality of life. This problem needs to be addressed by putting emphasis on not only the family but also the school and the school environment. All possibilities must be activated here that will encourage, above all, the physical stimulation of the body. In this case, we are not just talking about school physical education, but all extracurricular physical activities. 

The limitations of this study were certainly the number of probands observed during the second stage of data collection, which could be increased during future mapping. An issue associated with this fact involves considerable problems caused by the General Data Protection Regulation where school principals were unable to give us consent to conduct our research survey. The reason was the negative attitude of some parents, probably caused by fears of breaching the General Data Protection Regulation, despite it being clear that the research would be anonymous. Headteachers of some participating schools cited the General Data Protection Regulation as a reason for not participating in stage 2 of the research (after Covid-19). However, we believe that the main reason at the time was a concern about the potential risk of spreading Covid-19 and the isolation of schools from people other than teaching staff and essential school personnel. Simultaneously, we were not able to provide the coupling of data from the first and the second student´s survey due to the anonymity of the data. From the attached and completed chart, which characterise students from five selected schools during the first and second wave of the pandemic, it is obvious that the distribution of students during the first wave is similar to the general selection of students from all 15 schools. For that reason and in pursuit of achieving higher scientific and data validity, we implied the results from all 15 schools instead of the previous 5.

In our research survey, we used the BMI method of assessing overweight and obesity, which is advantageous when examining a larger sample of respondents because of the speed of the survey. BMI is a globally accepted index applicable to all age categories [16,36]. Indices calculated from anthropometric parameters, including BMI, should be assessed on the basis of national percentile charts and track the dynamics of their values over time. We are aware that the disadvantage of BMI is an inaccurate result for persons with a nonstandard proportion of some body tissue, such as an above-average proportion of muscle tissue. Such individuals were not represented in the survey. For a more accurate assessment of body composition and, therefore, obesity, it would be advantageous to use more accurate but more time-consuming methods, for example, calibration or bioelectrical impedance analysis [67].

## 5. Conclusions

In the research survey of a sample of 1456 upper primary school pupils in North Bohemia in the Czech Republic, we looked at physical activity and BMI before and after the COVID-19 disease. In assessing physical activity, we found a significant difference in the children who play sports regularly for at least 7 h per week (5%), children who attend sports playgroups with competitive training (8.74%), children who spend their free time inactively behind the computers for more than 14 h a week (5.4%), and the ones who spend their free time on social media for 8–14 h a week (18.65%). Simultaneously, we detected a significant difference among the children with normal weight (7.5%), children with stage 1 obesity (1.87%), and with the ones who are in the underweight category (6.24%).

It is clear from the results obtained by comparing groups of the children that there were statistically significant differences in the observed indicators of physical activity and BMI after the situation caused by the COVID-19 pandemic.

It should be pointed out that even before the adoption of antipandemic measures, i.e., during the collection of data in the first stage of the research, the children showed unsatisfactory results; 69% of the sample did not meet the recommended amount of physical activity, and only 67% of the sample had a normal weight. 

Given that physical activity is essential for the health of every individual, there needs to be a greater focus on changing social attitudes regarding inactivity and excess weight as a part of healthy lifestyle education. Conditions need to be created for young people to engage in sports and physical activities, especially in their leisure time. According to Roviello et al. [73], physical activity, especially outdoors, is beneficial to the fight against the current pandemic. We believe that the most important goal is to combine the interest of the family, school, and society with the interests and preferences of youth. This can be accomplished through everyday activities, by educating young people about sport and physical activity as the defining factors of a healthy lifestyle. This lifestyle is, among other things, about finding an active way to spend the leisure time that curbs the adverse effects of the social environment and the influence of modern technology, media, and advertising aimed at promoting questionable lifestyles.

## Figures and Tables

**Table 1 ijerph-19-03068-t001:** Distribution of boys and girls in the group according to the school year.

Grade	6th	7th	8th	9th	Total
Boys	219	195	179	182	775
%	29.42	25.22	22.42	22.94	
Girls	190	199	158	134	681
%	29.08	30.68	22.51	17.73	
Total	409	394	337	316	1456

**Table 2 ijerph-19-03068-t002:** Breakdown of pupils into categories by BMI before and after COVID-19 situation.

	Underweight	Normal Weight	Overweight	Class 1 Obesity	Class 2 Obesity	Total
1st stage *	217	761	131	22	2	1133
%	19.15	67.17	11.56	1.94	0.18	
2nd stage **	82	193	36	12	0	323
%	25.39	59.75	11.15	3.72	0.00	
Total	299	954	167	34	2	1456

Pearson Chi-square: 10.8593, *p* = 0.028198. *** before COVID-19, **** after COVID-19.

**Table 3 ijerph-19-03068-t003:** Pupils’ answers to the question “ Where do you do sports most often?” before and after the situation caused by COVID-19.

	A	B	C	D	E	Total
1st stage *	12	108	347	152	232	851
%	1.41	12.69	40.78	17.86	27.26	
2nd stage **	11	51	150	52	60	324
%	3.40	15.74	46.30	16.05	18.52	
Total	23	159	497	204	292	1175

Pearson Chi-square: 15.6897. *p* = 0.003467. Legend: A-exempt from PE, B-I play sports only at school, C-at school and exercise with friends, C-with family, D-at school and club-hobby club without competitive training, E-at school and competitive sports training. *** before COVID-19, **** after COVID-19.

**Table 4 ijerph-19-03068-t004:** Pupils’ answers to the question “How many hours a week do you do sports outside PE at school?” before and after situation caused by COVID-19.

	Doing Sports	Not Doing Sports	Total
1st stage *	257	582	839
%	30.63	69.37	
2nd stage **	83	240	323
%	25.70	74.30	
Total	340	822	1162

Pearson Chi-square: 2.74419, *p* = 0.097612, *** before COVID-19, **** after COVID-19.

**Table 5 ijerph-19-03068-t005:** Pupils’ responses to the question “Do you own a bicycle and use it as a means of transport or for sport?”, before and after situation caused by COVID-19.

	Yes	No	Total
1st stage *	765	86	851
%	89.89	10.11	
2nd stage **	277	47	324
%	85.49	14.51	
Total	1042	133	1175

Pearson Chi-square: 4.52669. *p* = 0.033373. *** before COVID-19, **** after COVID-19.

**Table 6 ijerph-19-03068-t006:** Pupils’ responses to the question “How many hours a week do you watch television?” before and after situation caused by COVID-19.

	0 to 3 h	4 to 7 h	8 to 14 h	14+ h	Total
1st stage *	476	242	104	28	850
%	56.00	28.47	12.24	3.29	
2nd stage **	180	99	37	8	324
%	55.56	30.56	11.42	2.47	
Total	656	341	141	36	1174

Pearson Chi-square: 1.00994, *p* = 0.798847, *** before COVID-19, **** after COVID-19.

**Table 7 ijerph-19-03068-t007:** Pupils’ responses to the question “How many hours a week do you work on the computer, play computer games or Play Station games?” before and after the situation caused by COVID-19.

	0 to 3 h	4 to 7 h	8 to 14 h	14+ h	Total
1st stage *	448	205	117	80	850
%	52.71	24.12	13.76	9.41	
2nd stage **	136	87	53	48	324
%	41.98	26.85	16.36	14.81	
Total	584	292	170	128	1174

Pearson Chi-square: 13.5056, *p* = 0.003663, *** before COVID-19, **** after COVID-19.

**Table 8 ijerph-19-03068-t008:** Pupils’ responses to the question “How many hours a week do you spend on your mobile phone or communicating on social media?” before and after situation caused by COVID-19.

	0 to 3 h	4 to 7 h	8 to 14 h	14+ h	Total
1st stage **	197	303	193	157	850
%	23.18	35.65	22.71	18.47	
2nd stage *	15	54	134	121	324
%	4.63	16.67	41.36	37.35	
Total	212	357	327	278	1174

Pearson Chi-square: 137.071. *p* = 0.0000001. *** before COVID-19, **** after COVID-19.

## Data Availability

The datasets generated during and/or analyzed during the current study are available from the corresponding author upon reasonable request.

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
