# Peer review of "Physical Activity and BMI before and after the Situation Caused by COVID-19 in Upper Primary School Pupils in the Czech Republic"

_ijerph, 2022, doi:10.3390/ijerph19053068_

Round 1
Reviewer 1 Report
This is an interesting work on the effects of COVID-19 pandemic on the inhabits of practicing physical activity, BMI and some worrying behaviours including spending long time in front of PC or with internet, within upper primary school pupils in Czech Republic.
The following revisions should be considered by authors:
-Abstract: provide more numeric values, e.g. for increase in obesity
-lines 55-64: check for consistency: civilization/civilisation
-Check if the english used in the entire manuscript is american or british english.
-line 114: better explain 'targeted mate selection'
-About 4 pages of Introduction are perhaps too many. Consider to delete some passages or moving to the Discussion, if relevant.
-the importance of physical activity, especially outdoors, is beneficial also in the fight against the current pandemic. Briefly mention it in the revised discussion or conclusions of the manuscript citing the work with DOI: 10.1007/s10311-021-01321-9
-Table 1: indicate 'grade' just once (up, left) and then report just numbers 6th, 7th, 8th and 9th ('th' should be upperscript)
-line 263-264 : rewrite 'The measurement of basic anthropometric indicators (body weight and height): Calibrated scale with accuracy'
-line 267 : rewrite 'Method of assessing body composition: From'
-line 268: 'to the square of the height in m' in place of "to height in m2'
-line 274 : 'PE ' explain here the meaning of this acronym
-Conclusions: 'Our research survey has shown that following the cessation
of the anti-pandemic measures, the observed values worsened even further in upper primary school children in North Bohemia.' Any hypothesis on the reason for this?
-Limitations: report them in a separated section.
-The authors report on problems caused to us by the General Data Protection Regulation (GDPR), causing the significant decrease of participants to the 2d stage of the study. Why didn't GDPR cause problems with the 1st stage? Please better clarify.
Reviewer 2 Report
This study attempts to explore the changes in physical activity and BMI before and after the situation 21 caused by the COVID-19 disease in upper primary school pupils in the Czech Republic. This topic is very meaningful, but there are some problems in this study.
1. The introduction and literature review are more about the role of sports on health, and are not closely related to the contents of this study.
2. The samples in the two stages are not the same group, and the sample number of the two stages varies greatly(1133 and 323), When comparing the samples of the two groups, it will be lack of scientificity..
3. The chi square test in table 2 ~ table 8 does not show which groups are different? Is it the difference between stage 1 and stage 2? Or the differences of different groups in same stage (such as the differences between groups A, B, C, D and E in stage 1 in Table 3)? However, the data only show one Pearson Chi Square value and P value, so it cannot clearly indicate the results described in the paper.
Round 2
Reviewer 2 Report
The data expression of chi square test in table 2-table 8 is unscientific. As shown in Table 2, the author shows that the difference test refers to the comparison between before covid-19 and after covid-19, but only one Pearson Chi Square: and P value is displayed. Which students this data refers to have significant differences? Is it the students who are under weight? Or the students who are normal weight? Or overweight? Class 1 obesity? Class 2 obesity? The data in the table should show whether there is a difference between before covid-19 and after covid-19 for the students who are under weight (and the other 4 groups). The same is for table 3-table 8.
Author Response
Dear Editor and Reviewer,
thank you for the factual comments. We revised the manuscript according to the referees' comments.
Author's Reply to the Review Report (Reviewer 2): The data expression of chi square test in table 2-table 8 is unscientific. As shown in Table 2, the author shows that the difference test refers to the comparison between before covid-19 and after covid-19, but only one Pearson Chi Square: and P value is displayed. Which students this data refers to have significant differences? Is it the students who are under weight? Or the students who are normal weight? Or overweight? Class 1 obesity? Class 2 obesity? The data in the table should show whether there is a difference between before covid-19 and after covid-19 for the students who are under weight (and the other 4 groups). The same is for table 3-table 8.
Author's Notes to Reviewer (after consulting with a leader in the field, with another statistician, prof. Cihlář): The chi-square test tests the null hypothesis that the two variables characteristic of the scale before covid and after covid are independent of each other. Therefore, the conclusion is for the whole table with the result that we reject this hypothesis based on the result (p=0.028198). If we were to ask whether there are significant differences in the percentages of children in each category (underweight, normal weight, overweight, obese stage I, obese stage II) we could visually compare the percentages listed in the first and second rows. However, the size of both samples should be taken into account. For this situation (comparison in only one category - for example, underweight) it would be possible to use the exact test of Difference between two proportions, but this was not the aim of our research. At the same time, for this test, we cannot obtain reliable results for the last three categories (overweight, obesity I and obesity II) because of their marginal representation. Since we wanted to compare the two periods (pre-covid and post-covid) as a whole, therefore chi-square test was used.
Best Regards,
Jana Pyšná et al.